

# Impact of Saharan dust on North Atlantic marine stratocumulus clouds: Importance of the semi-direct effect

Anahita Amiri-Farahani[1], Robert J. Allen[1], David Neubauer[2], and Ulrike Lohmann[2]

[1]University of California Riverside, Department of Earth sciences, Riverside, USA
[2]ETH Zurich, Institute for Atmospheric and Climate Science, Zurich, Switzerland

*Correspondence to:* Anahita Amiri-Farahani (aamir003@ucr.edu)

**Abstract.**

One component of aerosol-cloud interactions (ACI) involves dust and marine stratocumulus clouds (MSc). Few observational studies have focused on dust-MSc interactions, thus this effect remains poorly quantified. We use observations from multiple sensors in the NASA A-Train satellite constellation from 2004 to 2012 to obtain estimates of the aerosol-cloud radiative effect, including its uncertainty, for dust aerosol influencing Atlantic MSc off the coast of North Africa between 45°W and 15°E, and 0-35°N. To calculate the aerosol-cloud radiative effect, we use two methods following Quaas et al. (2008) (Method 1) and Chen et al. (2014) (Method 2). These two methods yield similar results of -3.99±0.78 and -3.21±3.61 $Wm^{-2}$, respectively, for the annual mean aerosol-cloud radiative effect. Thus, Saharan dust modifies MSc in a way that acts to cool the planet. There is a strong seasonal variation, with the aerosol-cloud radiative effect switching from significantly negative during the boreal summer to weakly positive during boreal winter. Method 1 (Method 2) yields -3.81 ±2.51 (-4.27±4.01) during summer, and 0.97 ±2.91 (0.63±0.48) $Wm^{-2}$ during winter. In Method 1, the aerosol-cloud radiative effect can be decomposed into two terms, one representing the first aerosol indirect effect and the second representing the combination of the second aerosol indirect effect and the semi-direct effect (i.e., changes in liquid water path and cloud fraction in response to changes in absorbing aerosols and local heating). The first aerosol indirect effect is relatively small, varying from -0.65±0.61 in summer to 0.05±0.5 $Wm^{-2}$ in winter. The second term, however, dominates the overall radiative effect, varying from -3.16 ±2.45 in summer to 0.92±2.86 $Wm^{-2}$ during winter. Studies show that the semi-direct effect can result in a negative (i.e., absorbing aerosol lies above low clouds like MSc) or positive (i.e., absorbing aerosol lies within low clouds) aerosol-cloud radiative effect. CALIPSO shows that 50% to 90% of Saharan dust resides above North Atlantic MSc during summer for most of our study area. This is consistent with a relatively weak first aerosol indirect effect, and also suggests the second aerosol indirect effect plus semi-direct effect (the second term in Method 1) is dominated by the semi-direct effect. In contrast, the percentage of Saharan dust above North Atlantic MSc is much lower during the winter, ranging from 10% to 40% . Because the aerosol-cloud radiative effect is positive during winter, and is also dominated by the second term, this again supports the importance of the semi-direct effect. We conclude that Saharan dust-MSc interactions off the coast of north Africa are likely dominated by the semi-direct effect.





# 1 Introduction

To reduce uncertainty in climate sensitivity and future global warming estimates, it is necessary to quantify the radiative forcing of aerosols. However there is a large uncertainty in aerosol radiative forcing, and much of this uncertainty is related to the magnitude of indirect aerosol effects on clouds of $-0.45$ $\mathrm{Wm}^{-2}$ with an uncertainty range of $-1.2$ to $0$ $\mathrm{Wm}^{-2}$ (Boucher et al., 2013). Aerosols also impact clouds through "rapid adjustments" associated with aerosol-radiation interactions, otherwise known as semi-direct effects (SDE). Available estimates suggest a relatively large SDE uncertainty of $-0.3$ to $+0.1$ $\mathrm{Wm}^{-2}$ (Boucher et al., 2013). The growing interest in the impact of aerosols on climate has stimulated the development of better physically based parameterizations of aerosols and aerosol-cloud interactions (ACI) in climate models. Nevertheless, the lack of understanding of external forcing on clouds remains one of the largest uncertainties in climate modeling and climate change projections.

One aspect of ACI is the possible influence of dust on marine stratocumulus (MSc) clouds. North Africa is the world's largest dust source (Goudie and Middleton, 2001). Dust emissions from this region occur from both the hyper-arid Sahara and the semi-arid Sahel. Africa is responsible for approximately half of the global emissions (Huneeus et al., 2011) with several hundred teragrams of dust being transported across the Atlantic towards the Americas throughout the year (Kaufman et al., 2005). This has consequences for air quality downwind (Prospero, 1999) as well as the radiative balance over the Atlantic, via scattering and absorption of solar radiation (and to a lesser extent absorption of terrestrial radiation), and microphysical and thermodynamical effects on clouds (Kaufman et al., 2005), and tropical cyclone formation (Evan et al., 2006). The dominant mode of coupled ocean-atmosphere variability in the tropical Atlantic is called Atlantic Meridional Mode (AMM). Evan et al. (2011) show that this mode is linked to Saharan dust variability. The AMM is thermally damped, thus direct ocean cooling from dust is required for the AMM to persist.

Along the western coast of Africa, extensive regions referred to as the semipermanent subtropical marine stratocumulus sheets exist, in which the stratocumulus cover exceeds $40\%$ and can be as high as $60\%$. Therefore, they may be affected by the high concentrations of continental aerosols, in particular dust. Stratocumulus clouds strongly reflect incoming solar radiation (Chen et al., 2000) but have a small effect on outgoing longwave radiation. Small changes in the coverage and thickness of stratocumuli are enough to produce a radiative effect comparable to that associated with increasing greenhouse gases (Randall et al., 1984; Slingo, 1990).

A few observational studies show a relation between dust aerosols and cloud cover. Mahowald and Kiehl (2003) show that there was a positive correlation between observed thin low cloud amount and mineral dust off the west coast of North Africa. Observations during a dust storm suggest smaller cloud droplets and suppressed precipitation over the eastern Mediterranean (Rosenfeld et al., 2001). In another study, rainfall and dust load in the West African Sahel exhibit a negative correlation, which is explained by a larger number of cloud condensation nuclei (CCN) when the dust load is high, distributing available cloud water over a large number of droplets, thus suppressing droplet growth and precipitation (Hui et al., 2008). Li et al. (2010) study the indirect effects of mineral dust on warm clouds during a Saharan dust-transport event. They show that clouds are affected strongly by dust and the effects segregate and vary systematically when classified by cloud precipitation regime,



cloud top temperature, and liquid water path (LWP). For nonprecipitating clouds the estimated aerosol indirect effect (AIE) is -0.07 $\mathrm{Wm^{-2}}$ over all temperature bands. Further classification by LWP (for all LWP > $150\mathrm{gm^{-2}}$) strengthens the AIE to approximately -0.16 $\mathrm{Wm^{-2}}$. McComiskey et al. (2009) present an assessment of ACI from ground-based remote sensing under coastal stratiform clouds. They calculate ACI as the change in cloud droplet number concentration ($N_d$) with aerosol

concentration for constant values of LWP. They show that the average ACI depends on the relative value of cloud LWP, methods for retrieving $N_d$, the aerosol size distribution, updraft velocity, and the scale and resolution of observations. Doherty and Evan (2014) show that over the tropical North Atlantic during summer, low cloud fraction increases by 3-10% in response to high mineral dust loadings.

In this paper, we will show the importance of Saharan dust contributions to ACI off the coast of North Africa, and in

particular, the importance of the SDE. Initial modeling studies found that the SDE causes a positive radiative forcing, thus warming the climate system (Hansen et al., 1997; Allen and Sherwood, 2010). Furthermore, Ackerman et al. (2000) show that when absorbing aerosol coincides with shallow broken clouds, the radiative heating of absorbing aerosol reduces the cloud cover and increases the absorption solar radiation at the surface, resulting in a net positive radiative forcing. However, more recent modeling studies show that when absorbing aerosol resides above the cloud top, it can stabilize the underlying layer,

enhancing stratocumulus clouds (Koch and Del Genio, 2010; Allen and Sherwood, 2010). Johnson et al. (2004), using large eddy simulation experiments, show absorbing aerosols may also yield increased cloud cover and surface cooling under certain scenarios. Although few observational studies exist to corroborate these model results, Wilcox (2010) uses satellite data and shows that when smoke resides above stratocumulus clouds, the increased buoyancy of the air above the clouds inhibits the entrainment of dry air, which helps preserve humidity and cloud cover in the boundary layer. Similarly, Brioude et al. (2009)

showed the overall effect of biomass burning was to enhance marine stratocumulus off the coast of California. Koren et al. (2004), however, show that Amazonian biomass burning suppressed satellite-based cumulus cloud cover.

Here we quantify the radiative effects of Saharan dust on North Atlantic MSc. We use observations from multiple sensors in the NASA A-Train satellite constellation from 2004 to 2012 to evaluate the complex processes inherent in aerosol-cloud systems and to obtain estimates of aerosol-cloud radiative forcing for dust and marine stratocumulus clouds, including the un-

certainties. The NASA data include CloudSat radar observations co-located with aerosol and cloud properties from CALIPSO, CERES and ERA-Interim reanalysis data. We show that the SDE−relative to the first and second aerosol indirect effects−is the largest component of ACI, and is also responsible for a seasonal reversal in the sign of ACI. A description of our datasets and methodology are provided in Sections 2 and 3. Results are presented in Section 4, and a discussion/conclusion is presented in Section 5.

## 2  Data

Cloud-Aerosol Lidar with Orthogonal Polarization (CALIOP) instrument on board the Cloud-Aerosol Lidar and Infrared Pathfinder Satellite Observation (CALIPSO; Winker et al. 2009) has provided data since June 2006. This space lidar measures the backscatter signal at 532 and 1064 nm and the degree of linear polarization at 532 nm. CALIOP provides aerosol and



cloud profiles with high vertical resolution of 30-60 m (up to 20 km) during its 16-day repeat cycle, and its beam diameter is 70 m at the surface (Winker et al., 2007). CALIOP has a very small swath width and the distance between two CALIPSO tracks is more than 2000 km in low and mid-latitudes. Thus, to produce statistically meaningful profiles, a significant averaging in time and space is needed (Winker et al., 2010).

CALIOP can discriminate between dust and other types of aerosols, which generally do not depolarize light. Due to CALIOP's sensitivity to polarization at 532 nm, the depolarization from scattering from non-spherical dust particles is a means to discriminate between dust and other aerosol species (Amiridis et al., 2013). CALIPSO categorizes aerosols into six sub-categories: dust, marine, smoke, polluted dust, polluted continental, and clean continental (Young and Vaughan, 2009). Compared to the Moderate-Resolution Imaging Spectrodiometer (MODIS) sensor, most studies show that CALIPSO underes-

timates dust aerosol optical depth (DAOD) of the order of 0.1 over the regions having strong mineral dust load (e.g., Redemann et al. (2012)). Amiridis et al. (2013) demonstrate improvements in CALIPSO dust extinction retrievals over northern Africa and Europe. The improvement is applied by corrections to the Saharan dust lidar ratio assumption for CALIPSO level2 data, the 5 km aerosol layer product (version 3.01), and separation of the dust portion in detected dust mixtures, and the averaging scheme introduced in the CALIPSO level 3 product. For this study dust vertical profiles are obtained from Amiridis et al.

(2013). CALIPSO gives extinction coefficient of dust for 399 vertical levels. DAOD at each level is calculated as the vertical integral of dust extinction profile at 532 nm. By using CALIPSO it is possible to quantify how much dust is above clouds and how much is within or below clouds. CALIPSO data is available from 2007 to 2014 for this study.

CALIPSO gives only two or three DAOD values per month per grid box, thus it is not possible to use daily CALIPSO to infer statistical relationships. Daily DAOD is obtained from Monitoring Atmospheric Composition and Climate (MACC)

reanalysis. The MACC global reanalysis consists of a long-term reanalysis (2003-2012) with the coupled MACC system with data assimilation of aerosol optical depth (AOD) from MODIS satellite data. Different aerosol species (sea salt, dust, organic matter, black carbon, and sulphate) are included in MACC (Inness et al., 2013). AOD is also obtained from MACC at 0.55 $\mu m$ and 0.865 $\mu m$ wavelengths.

The Clouds and the Earth's Radiant Energy System (CERES) (Wielicki et al., 1996; Loeb et al., 2005, 2007) products

include both solar-reflected and terrestrial radiation from the top of the atmosphere to the Earth's surface. Daily data of cloud properties such as effective cloud-particle radius ($r_e$), cloud optical thickness, cloud cover and liquid water path (LWP) are from the CERES Aqua Single Scanner Footprint (SSF) Edition 3A data set. Daily values of clear sky albedo from 2004 to 2012 are also derived from CERES for this study. All satellite data are obtained on a $1° \times 1°$ resolution.

The MSc regime is defined by lower-tropospheric stability (LTS) and vertical velocity. To calculate potential temperature,

daily temperature is obtained from the ERA-Interim (Dee et al., 2011) reanalysis at 1000 hPa and 700 hPa levels at $1° \times 1°$ resolution. Daily mean vertical velocity at 500 hPa is also obtained from ERA-Interim.





## 3 Method

### 3.1 Study Area

Our study area is the tropical North Atlantic, defined between 45°W and 15°E, and 0-35°N. The boundaries of our study area are based on the location of the MSc regime and high dust load over the North Atlantic Ocean. Figure 1 shows DAOD from

5 MACC for different seasons. During winter (December-January-February), dust is found within 0-15 °N off western Africa, over the North Atlantic Ocean. In summer (June-July-August), dust moves farther northward, occurring off the western coast of Africa between 10-25°N. During spring (March-April-May) and fall (September-October-November), dust is located between its wintertime and summertime locations. The maximum westward dust transport, as well as the maximum dust loading, occur during summer, with relatively high dust load out to ∼45°W.

A cloud regime based analysis is used to identify marine stratocumulus clouds (Medeiros and Stevens, 2011). The MSc regime is defined as 500 hPa vertical velocity $> 10 hPa\ day^{-1}$ and to separate trade-wind cumuli from MSc, a LTS criterion is used, defined as $LTS = \Theta_{700hPa} - \Theta_{1000hPa} > 18.55K$ (where $\Theta$ is the potential temperature). Retrievals over bright surfaces like deserts are unreliable, so land areas are excluded. Figure 2 shows the percent of days in which the stratocumulus regime exists. During summer, between 10-40°N and 10-45° W, MSc occur from 50% to 80% of the days. The percent of days the

MSc regime occurs, is lower during the other seasons−particularly during fall−but the location is similar.

### 3.2 Satellite Methodology

Rosenfeld et al. (2014) show that when $r_e$ reaches about 14 $\mu m$ the coalescence accelerates and initiates warm rain. We only focus on non-raining clouds ($r_e <$ 14 $\mu m$), because under raining conditions, the relationship between cloud properties and DAOD may be subject to aerosol removal by precipitation and thus more difficult to analyze directly. Following Quaas et al.

(2006), thin clouds with cloud optical thickness less than 4 and cloud effective radius less than 4 $\mu m$ are excluded since neither a clear distinction between aerosols and clouds, nor an accurate retrieval of cloud properties is reliable in such cases.

$N_d$ is estimated using the adiabatic approximation (Brenguier et al., 2000). This relationship assumes that liquid water content and cloud droplet radius increase monotonically with height in the cloud with a constant $N_d$ in the vertical. Hence, $N_d$ can be computed from cloud optical depth and $r_e$:

$$N_d = \gamma \tau_c^{1/2} r_e^{-5/2} \tag{1}$$

where $\tau_c$ is cloud optical depth, with $\gamma = 1.37 \times 10^{-5} m^{-0.5}$ (Quaas et al., 2006). Table 1 shows variables with their definitions used in the equations. Quaas et al. (2008) show that the planetary albedo ($\alpha$) is described by contributions of clear and cloudy parts of the scene. They use a combination of CERES and MODIS products for a sigmoidal fit to describe the albedo





of a cloudy scene involving liquid water clouds and extend it to include the clear part of the scene, where the planetary albedo also depends on the AOD. We use this approach to define the planetary albedo:

$$\alpha \approx (1-f)[a_1 + a_2 \ln \tau_a] + f_{liq}[a_3 + a_4(f\tau_c)^{a_5}]^{a_6} + f_{ice}\alpha^{icecld} \tag{2}$$

where $\tau_a$ is aerosol optical depth, $f$ is the fraction of all clouds including both liquid water and ice clouds ($f = f_{liq} + f_{ice}$), and $\alpha^{icecld}$ is the planetary albedo for the parts covered by ice clouds. $a_1$ - $a_6$ are fitting parameters taken from Quaas et al. (2008). The first term on the right hand side of this expression refers to planetary albedo in the clear sky and the second term describes the cloudy parts of the scene. The last term shows the contribution of ice clouds to the planetary albedo. Since we are interested in the effect of dust on MSc (which are warm clouds), $f=f_{liq}$ in this study and the last term can be neglected.

Aerosol index (AI= aerosol optical depth × Ängström exponent) is derived from MACC and is used as a proxy for column CCN. The Ängström parameter is defined as:

$$\beta = -\frac{\ln(\frac{AOD_{\lambda_1}}{AOD_{\lambda_2}})}{\ln(\frac{\lambda_1}{\lambda_2})} \tag{3}$$

The Ängström exponent is calculated on the basis of AOD at 0.55 $\mu m$ and 0.865 $\mu m$ (Remer et al., 2005). It provides information on the particle size; the larger the exponent, the smaller the average size of the particles. The AI gives lower weight to large aerosols and reduces the impact of large but low number-concentration sea salt and dust particles (Stier, 2016). Liu and Li (2014) find improved correlation between surface CCN and AI as compared to AOD. Figure 3 shows the spatial pattern of the Ängström exponent for different seasons. It has smaller values over North Africa and the neighboring ocean, indicating larger particles (dust) reside there.

To estimate the aerosol cloud radiative effect, statistical relationships between dust and clouds are calculated, following Quaas et al. (2008) (Method 1) and Chen et al. (2014) (Method 2) respectively. In Method 1, the radiative effect is decomposed into the first AIE and the combination of the second aerosol indirect effect (the cloud lifetime effect (CLE)) and the semi-direct effect. The first aerosol indirect radiative effect, or the cloud albedo effect, is calculated as the change in $N_d$ to the change in AI:

$$AIE = f \cdot A(f, \tau_c)\frac{1}{3}\frac{d \ln N_d}{d \ln(AI)}[\ln \tau_a - \ln(\tau_a - \tau_{dust})]\overline{F} \downarrow \tag{4}$$

The second part corresponds to the combination of the CLE and the SDE, and includes both changes in LWP and cloud fraction to the change in AI:





$$CLE + SDE = [(\alpha - (a_1 + a_2 \ln \tau_a)) \frac{d \ln f}{d \ln(AI)}$$
$$+ f \cdot A(f, \tau_c)(\frac{d \ln f}{d \ln(AI)} + \frac{d \ln(LWP)}{d \ln(AI)})]$$
$$[\ln \tau_a - \ln(\tau_a - \tau_{dust})]\overline{F} \downarrow \qquad (5)$$

where $f$ is the marine stratocumulus cloud coverage and clouds are not obscured by overlying ice clouds. $\overline{F} \downarrow$ is the mean

daily downward solar radiation flux at the top of the atmosphere, in $\mathrm{Wm^{-2}}$, as a function of the latitude and the day of the year. $\alpha$ is the planetary albedo, $N_d$ is the liquid $N_d$, $\tau_a$ is the AOD and $\tau_{dust}$ is the DAOD. A detailed description of the computation of equations (3-5), and $A(f, \tau_c)$ are given in the Appendix of Quaas et al. (2008).

In Method 2 the aerosol radiative effect includes the intrinsic effect (i.e., aerosol variations on cloud albedo, the combination of changes in cloud droplet size and LWP on cloud albedo) and the extrinsic effect (i.e., aerosol variations on fractional cloud

cover). The aerosol radiative effect is calculated as the change in clear sky and cloud albedo, to the change in AI plus the change of cloud fraction to a change in AI:

$$RF = [\overline{C_m}(\frac{dA_{clr}}{d \ln(AI)} - \frac{dA_{cld}}{d \ln(AI)}) + (\overline{A_{clr} - A_{cld}}) \frac{df_{cld}}{d \ln(AI)}][\ln \tau_a - \ln(\tau_a - \tau_{dust})]\overline{F} \downarrow \qquad (6)$$

Where $\overline{C_m}$ is the seasonal mean marine stratocumulus cloud coverage, $A_{clr}$ is clear-sky albedo, and $A_{cld}$ is the cloudy-sky albedo. The cloudy-sky albedo is derived using:

$A_{cld} = [\alpha - (1 - f)A_{clr}]/f \qquad (7)$

The first and second term on the right hand side of eq. (6) are called the intrinsic and extrinsic effect respectively. Method 2 is an alternative way to estimate the total radiative effect which can be compared to Method 1. Contrary to Method 1, it is not possible to decompose the total aerosol-cloud radiative effect into the AIE and the combination of CLE and SDE. Thus we compare the total aerosol radiative effect estimated by these two methods.

To estimate the aerosol cloud radiative effect, linear regressions of each partial derivative are calculated. Each data point in the regression represents a day for which both dust and MSc data exist for the grid point. The sensitivities and radiative effects are calculated on a $1° \times 1°$ grid. In both methods, sensitivities with fewer than ten contributing data points are excluded. The uncertainty is computed from one-sigma error of the linear regression fit.

Gryspeerdt et al. (2016) show that by including information about $N_d$, the impact of the meteorological covariations in the

susceptibility analysis is significantly reduced and much of the correlation between AOD and cloud fraction is explained by



other factors than that mediated by $N_d$. They show that by considering these, the strength of the global mean relationship of AOD and cloud fraction is reduced by around 80%. We follow their new method and calculate this relationship as follows:

$$\frac{df}{d\ln(AI)} = \frac{df}{d\ln N_d} \cdot \frac{d\ln N_d}{d\ln(AI)} \tag{8}$$

## 4  Results

Here we present the annual and seasonal radiative effect of dust on MSc, as estimated by both Method 1 and Method 2. The annual mean aerosol cloud radiative effect estimated by Method 1 is -3.99±0.78 $\mathrm{Wm}^{-2}$ (Table 2). The negative radiative effect indicates that dust modifies MSc in a way that results in a cooling effect over the study area. Method 1 separates the aerosol cloud radiative effect into two terms (Equations 4 and 5). Figure 4 shows the first aerosol indirect effect for different seasons. In all figures white areas indicate missing values, where no data for dust or cloud exist, or insufficient data exists to calculate

the partial derivatives.

The first indirect effect is stronger where the dust load is larger and the stratocumulus regime exists for a longer time (see Figures 1 and 2). The annual mean first indirect effect is -0.69±0.29 $\mathrm{Wm}^{-2}$, and it varies from -0.65±0.61 $\mathrm{Wm}^{-2}$ in summer to 0.05±0.5 $\mathrm{Wm}^{-2}$ in winter (Table 2). The larger negative radiative effect during summer, compared to spring and fall, is consistent with a greater abundance of both MSc and dust during summer.

Figure 5 shows the combination of the CLE and SDE (i.e., the second term in Method 1). Similar to the cloud albedo effect, the CLE + SDE is negative during summer, fall and spring and positive during winter. Moreover, CLE + SDE also exhibits a summertime maximum (negative), which is again consistent with the greater abundance of MSc and dust during summer. For all seasons the second term is much larger than the first term. The second term varies from -3.16±2.45 $\mathrm{Wm}^{-2}$ in summer to 0.92±2.86 $\mathrm{Wm}^{-2}$ in winter, with an annual mean of -3.3±1.09 $\mathrm{Wm}^{-2}$. This shows the importance of CLE and SDE in the

study area.

Method 2 yields similar conclusions on the magnitude of the total aerosol cloud radiative effect, as well as the seasonal variation. The annual mean aerosol cloud radiative effect for Method 2 is -3.21±3.61 (Table 2), and it varies from -4.27±4.01 in summer to 0.63±0.48 in winter. Method 2 separates the radiative effect into intrinsic and extrinsic parts, which are shown in Figure 6 and 7, respectively. The intrinsic effect dominates the radiative effect in this method. Like Method 1 the radiative

effect is more negative over areas with larger dust load and a higher percentage of days with MSc.

In Method 1, CLE and SDE dominate the total aerosol-cloud radiative effect. To investigate the role of the SDE over the region, we look at the vertical profile of Saharan dust during winter and summer from CALIPSO. Figure 8 shows that during winter, most dust burden resides between 0-1 km. In contrast, during summer there are two peaks in Saharan dust: one peak is within the marine boundary layer (between 0-1km), and the rest resides above the boundary layer. Since MSc reside within

the boundary layer, a considerable amount of dust resides above the clouds during summer. We use cloud top height for those days where the vertical profile of dust extinction coefficient from CALIPSO is available and calculate how much dust is above the top of MSc. Figure 9 shows that 50-90% of the dust resides above MSc for most of our study area during the summer;





only 10-40% resides above MSc during the winter. Tsamalis et al. (2013) show that during the summer, the Saharan air layer is found to be thicker and higher near Africa at 1-5 km. During winter, it occurs in the altitude range 0-3 km off the western Africa. This is consistent with the vertical profile of Saharan dust in our study. This vertical profile analysis helps to explain the relatively weak first term of Method 1, relative to the second term. It also implies the second term is dominated by the SDE.

To investigate this more, we plot the two partial derivatives that constitute the second term of Method 1. Figure 10 and 11 show the sensitivity of the cloud fraction and LWP to a relative change in AI for all seasons. Figure 10 shows that the sensitivity of cloud fraction to AI is relatively weak. It also shows that this sensitivity is positive (negative) during summer (winter) for most of the study area, which shows that cloud fraction increases (decreases) when AI increases. Figure 11 shows that the sensitivity of LWP to AI dominates the second term of Method 1. During winter, most of the study area features a reduction in
LWP with respect to the AI. During summer, however, this sensitivity is generally positive. Considering the seasonal contrast in the amount of dust above MSc during summer versus winter, the seasonal reversal of these sensitivities−which drive the reversal in the total aerosol cloud radiative effect−is consistent with the importance of the SDE.

Since the bulk of the dust resides above MSc during summer, aerosol-cloud microphysical interactions (including AIE and CLE) would be muted. Thus, AIE and CLE would be smaller than SDE. Moreover the SDE would be negative, as observed
by the CLE+SDE term of Method 1. Wilcox (2010) also shows that absorbing aerosols overlying MSc largely do not interact with the clouds. However, the aerosols still result in cloud thickening by a dynamical feedback related to the enhanced stability of the atmosphere, which yields an increase in the cloud albedo. This is consistent with Koch and Del Genio (2010), who show absorbing aerosol above MSc result in increased stability, which strengthens the inversion, and reduces cloud-top entrainment of overlaying dry air, thereby enhancing the underlying clouds. Evan and Doherty (2014) show that in response to increased
dust load over the tropical North Atlantic in summer, MSc also increase, and this is linked to increases in atmospheric stability, reductions in boundary layer height, and moistening of the lower atmosphere.

During winter, when the total aerosol cloud radiative effect reverses sign and becomes positive, most of dust burden resides within or below the clouds. When absorbing aerosol coincides with the cloud, the heating favors cloud clearing and thinning, thus reducing the cloud albedo and yielding a positive radiative effect (Hansen et al., 1997; Johnson et al., 2004). In contrast,
aerosol indirect effects do not drive cloud clearing/thinning, and thus do not contribute a positive radiative effect. Therefore, over our study area, we conclude that the SDE is the most important aerosol-cloud effect resulting in an overall negative radiative effect. The SDE is also strong enough to change the sign of total aerosol cloud radiative effect from negative to positive during the winter.

## 5   Conclusions

To estimate the aerosol-cloud radiative effect of Saharan dust on North Atlantic MSc , we use observational data from several different satellites from 2004 to 2012. The aerosol-cloud radiative effect is estimated using two different methods, following Quaas et al. (2008) (Method 1) and Chen et al. (2014) (Method 2). The annual mean aerosol cloud radiative effect estimated by Method 1 is -3.99±0.78Wm$^{-2}$. Estimating the radiative effect using Method 2 yields similar results, with an annual mean of -





$3.21\pm3.61$ Wm$^{-2}$. Thus, both methods show that Saharan dust modifies MSc in a way that has a cooling effect over the North Atlantic Ocean. Both methods also yield a seasonal maximum negative radiative effect during summer, which is consistent with more Saharan dust and MSc during summer. Furthermore, both methods yield a reversal in the sign of the aerosol cloud radiative effect, which switches from negative to positive during the winter season. In Method 1, the radiative effect varies from

5    -3.81$\pm$2.51 Wm$^{-2}$ during summer to 0.97$\pm$2.91 Wm$^{-2}$ during winter; similarly, Method 2 varies from -4.27$\pm$4.01 during summer to 0.63$\pm$0.48 during winter.

Method 1 allows us to separate the cloud albedo effect (first term of Method 1) from the CLE and the SDE (second term of Method 1). The cloud albedo effect, which varies from -0.65$\pm$0.61 Wm$^{-2}$ in summer to 0.05$\pm$0.5 Wm$^{-2}$ in winter and is relatively small compared to the CLE+SDE, which varies from -3.16 $\pm$2.45 Wm$^{-2}$ in summer to 0.92$\pm$2.86 Wm$^{-2}$ during

winter. This shows the importance of the second term, the combination of the CLE and the SDE.

To gain insight as to whether CLE or SDE dominates the second tern of Method 1, we use CALIPSO data to quantify the amount of Saharan dust that resides above MSc. The analysis shows that about 50-90% of Saharan dust resides above MSc during summer, but only around 10-40% resides above MSc during winter. This seasonal dependence in the location of the dust, relative to MSc, shows the importance of the SDE.

When most dust resides above the clouds during summer, aerosol-cloud microphysical effects that involve the co-location of aerosol and cloud, such as the second aerosol indirect effect (CLE), would likely be muted relative to the SDE. Moreover, the positive value of the aerosol-cloud radiative effect during winter, when most dust resides within MSc, indicates that the SDE is dominant$-$ that is the only mechanism by a negative aerosol-cloud radiative effect can be obtained. We conclude that aerosol-cloud radiative effects associated with Saharan dust and North Atlantic MSc are dominated by the semi-direct effect.

*Acknowledgements.* This study was funded by NSF award AGS-1455682 and a doctoral exchange grant through the Zeno Karl Schindler Foundation, which allowed Anahita Amiri-Farahani to complete part of this project at ETH Zurich. Authors would like to thank Vassilis Amiridis and Eleni Marinou for providing CALIPSO dust data and Johannes Quaas for giving the planetary albedo data.





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





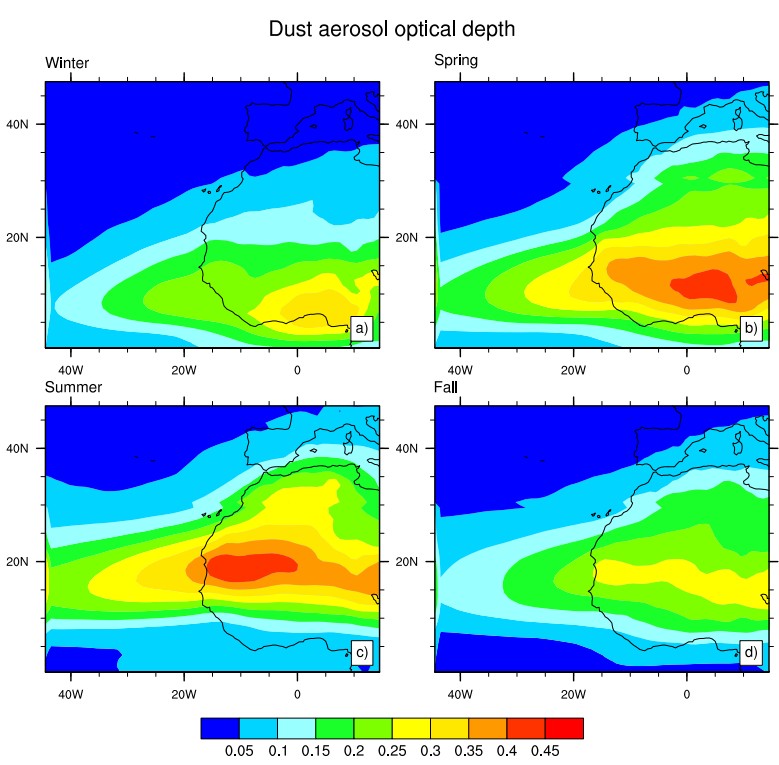

**Figure 1.** MACC dust aerosol optical depth (DAOD) from 2004-2012 in (a) winter, (b) spring, (c) summer, and (d) fall.





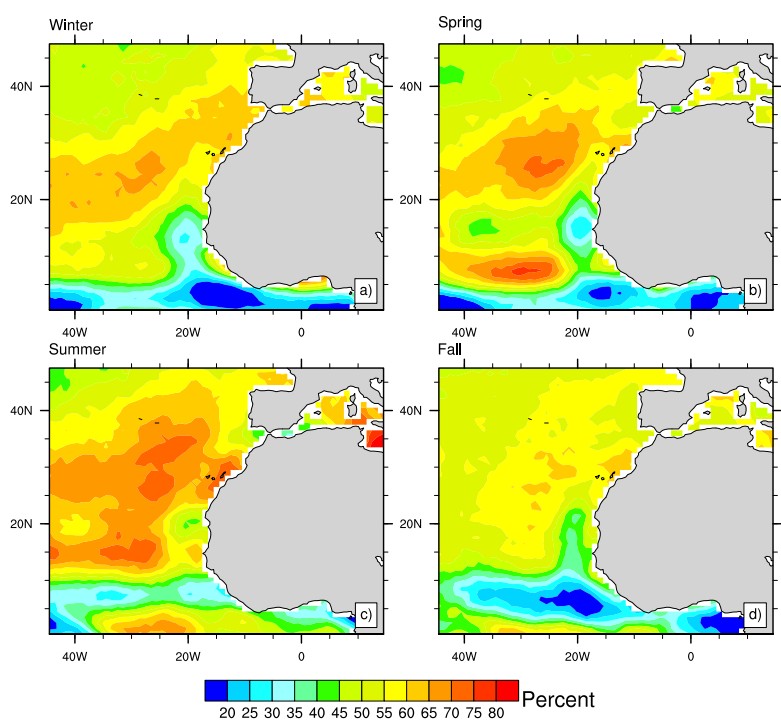

**Figure 2.** Percent of days from 2004-2012 in which marine stratocumulus clouds are found following Medeiros and Stevens (2011) in (a) winter, (b) spring, (c) summer, and (d) fall.





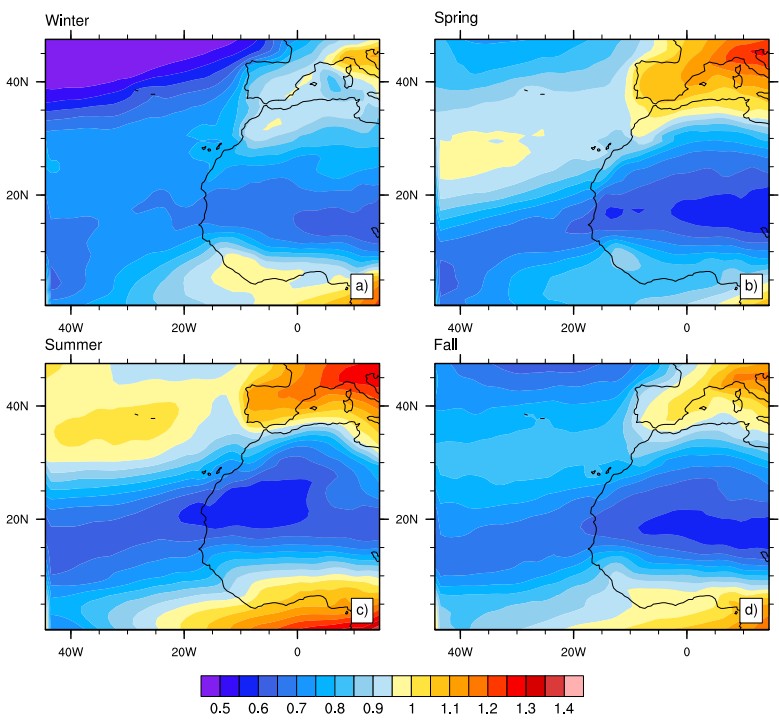

**Figure 3.** Aerosol Ängström exponent from MACC in (a) winter, (b) spring, (c) summer, and (d) fall.





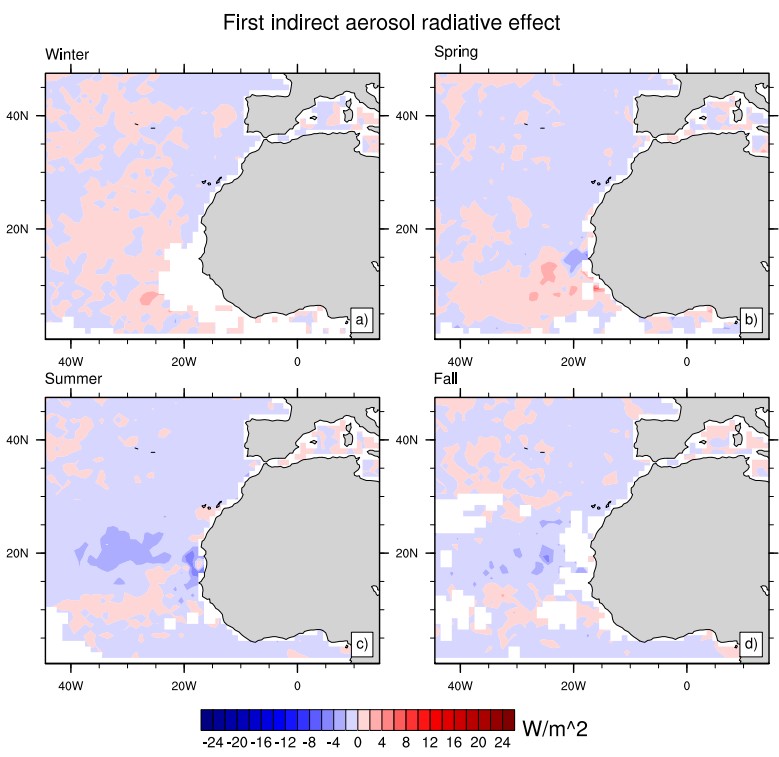

**Figure 4.** Fist indirect radiative effect (cloud albedo effect) of dust on marine stratocumulus clouds ($Wm^{-2}$) following Quaas et al. (2008) for (a) winter, (b) spring, (c) summer, (d) fall.





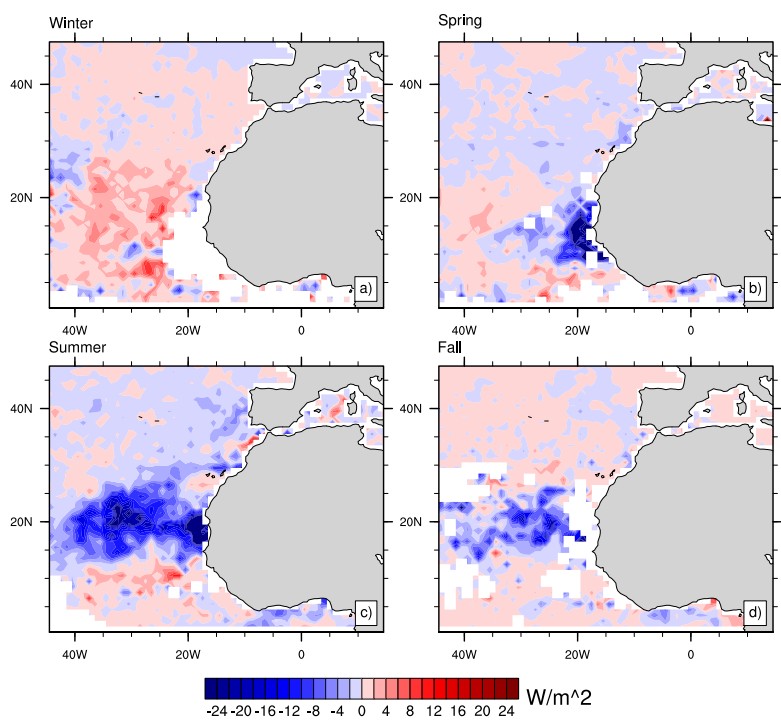

**Figure 5.** The second term of Method 1 (Quaas et al., 2008), which represents the cloud lifetime effect and semi-direct effect of dust on marine stratocumulus clouds ($Wm^{-2}$) which includes CLE+SDE for (a) winter, (b) spring, (c) summer, (d) fall.





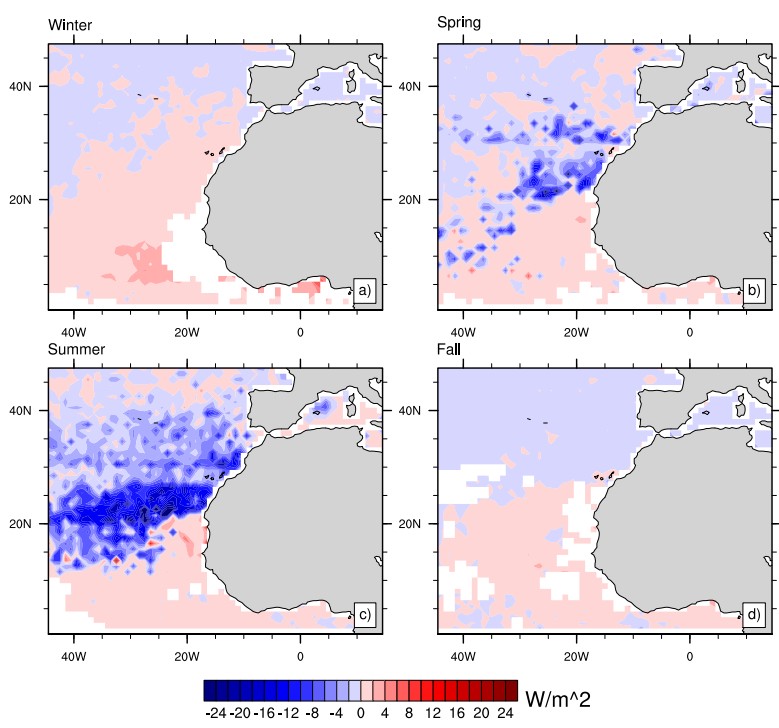

**Figure 6.** The intrinsic aerosol cloud radiative effect estimated for marine stratocumulus clouds ($Wm^{-2}$) following Chen et al. (2014) for ($Wm^{-2}$) for (a) winter, (b) spring, (c) summer, (d) fall.



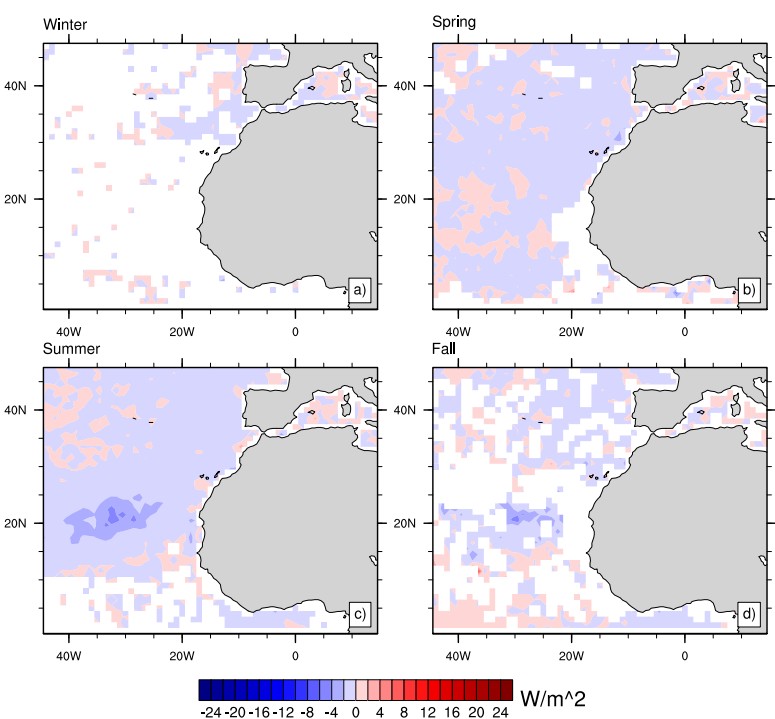

**Figure 7.** The extrinsic aerosol cloud radiative effect ($Wm^{-2}$) following Chen et al. (2014) for (a) winter, (b) spring, (c) summer, (d) fall.





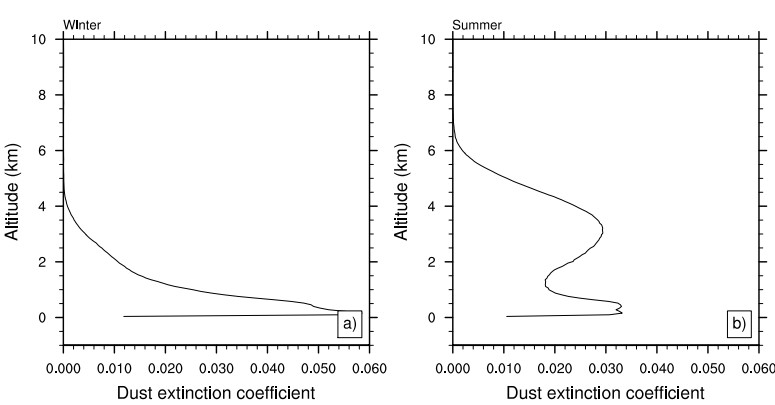

**Figure 8.** Vertical profile of dust from CALIPSO in (a) winter, (b) summer.





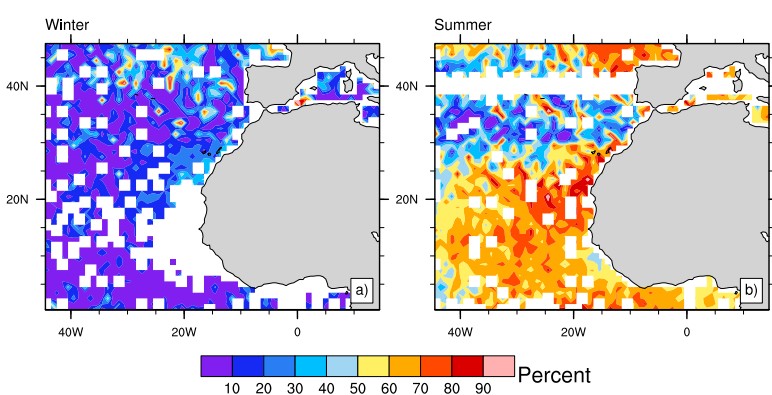

**Figure 9.** Amount of dust (%) above marine stratocumulus clouds in (a) winter and (b) summer.





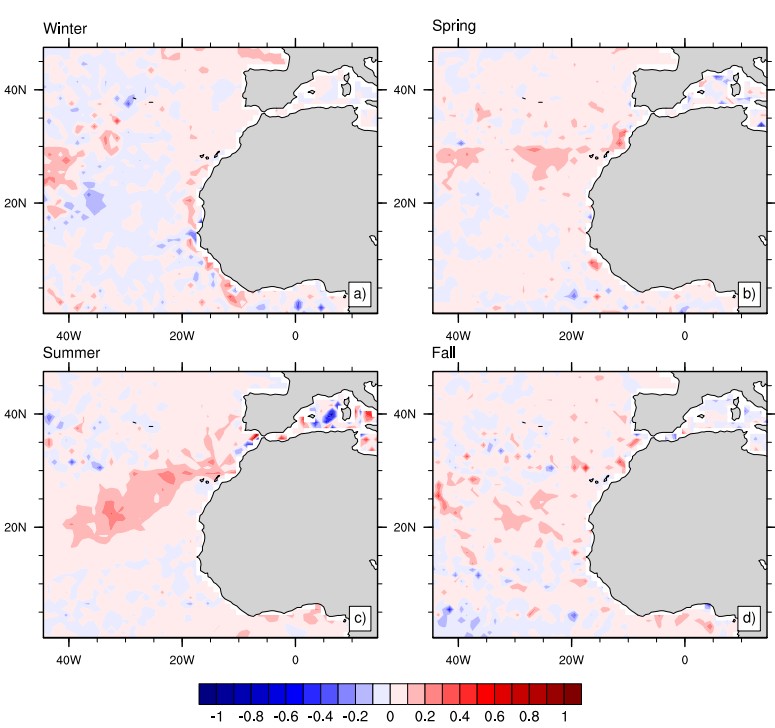

**Figure 10.** The sensitivity of cloud fraction to a relative change in aerosol index for (a) winter, (b) spring, (c) summer, (d) fall.





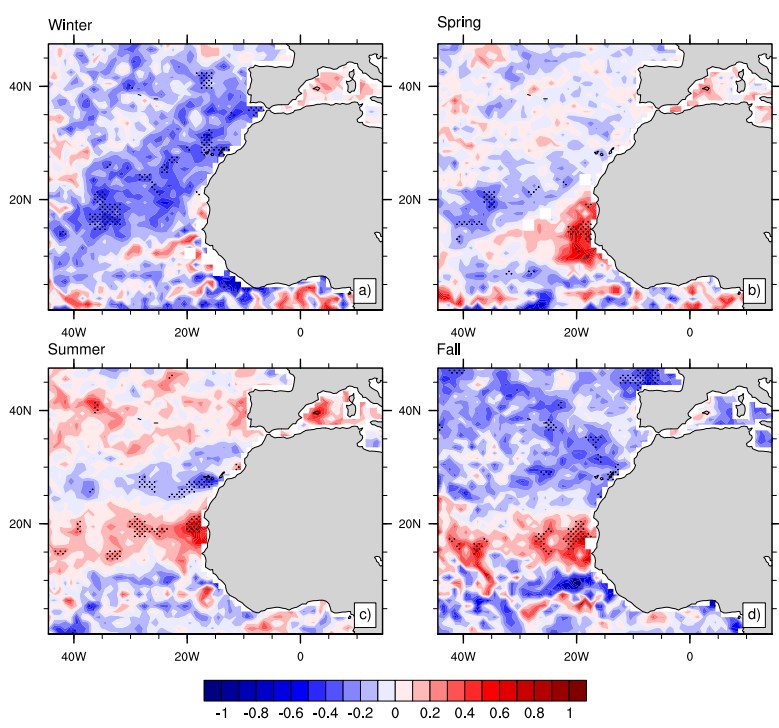

**Figure 11.** The sensitivity of liquid water path to a relative change in aerosol index for (a) winter, (b) spring, (c) summer, (d) fall. Dots represent the significance at 95% confidence level.



**Table 1.** A summary of notation used for equations in this paper

| Symbol | Meaning |
|---|---|
| $N_d$ | Cloud droplet number concentrations |
| $\alpha$ | Planetary albedo |
| $f$ | Total cloud fraction including both liquid water and ice cloud fraction |
| $f_{ice}$ | Ice cloud fraction |
| $f_{liq}$ | Liquid water cloud fraction |
| $\tau_a$ | Aerosol optical depth |
| $\tau_c$ | Cloud optical depth |
| $\tau_{dust}$ | Dust aerosol optical depth |
| $r_e$ | Effective cloud-particle radius |
| $LWP$ | Liquid water path |
| $A_{cle}$ | Clear-sky albedo |
| $A_{cld}$ | Cloudy-sky albedo |
| $\overline{C_m}$ | Seasonal mean MSc |
| $\bar{F}\downarrow$ | daily mean solar radiation at TOA |
| $AI$ | Aerosol Index |

**Table 2.** Seasonal and annual radiative effects estimated by Method 1 (Quaas et al., 2008) and Method 2 (Chen et al., 2014).

| | Method 1 | | | Method 2 |
|---|---|---|---|---|
| | AIE | CLE+SDE | Total radiative Effect | Total radiative Effect |
| Winter | 0.05±0.50 | 0.92±2.86 | 0.97±2.91 | 0.63±0.48 |
| Spring | -0.03±0.85 | -1.38±3.1 | -1.41±3.15 | -1.31±3.97 |
| Summer | -0.65±0.61 | -3.16±2.45 | -3.81±2.51 | -4.27±4.01 |
| Fall | -0.38±0.48 | -1.2±2.36 | -1.58±2.4 | -0.95±1.94 |
| Annual | -0.69±0.29 | -3.30±1.09 | -3.99±0.78 | -3.21±3.61 |