# Peer review of "Impact of Saharan dust on North Atlantic marine stratocumulus clouds: Importance of the semi-direct effect"

_Atmospheric Chemistry and Physics, 2016_

## Short Comment (SC1) · 9 Nov 2016

There is no reference to papers by Kishcha et al. (2014, 2015) on significant cloud cover up to 0.8 – 0.9 (created by desert dust intrusions) along the Saharan Air Layer (SAL) in the tropical Atlantic.

In the Abstract, the authors (Amiri-Farahani et al., ACPD, 2016) highlight that: "Few observational studies have focused on dust-MSc interactions, thus this effect remains poorly quantified". Unfortunately, the authors missed our publications on the topic (Kishcha et al., 2014; 2015). Based on MODIS cloud fraction data and NASA MER-RAero aerosol reanalysis data during a 10-year period from July 2002 to June 2012, Kishcha et al. (2014, 2015) found that, in July, dust intrusions from the Sahara into

the tropical Atlantic contribute to significant cloud cover up to 0.8 − 0.9 along SAL. The area of SAL with significant CF is characterized by limited precipitation, indicating that clouds along the SAL are not developed enough. To explain the observed significant cloud cover up to 0.8 − 0.9 along SAL, Kishcha et al. (2015) suggested a plausible physical mechanism based on the indirect effect of Saharan dust on stratocumulus clouds below the temperature inversion under the base of SAL. Based on MODIS-derived effective radius of cloud droplets, Kishcha et al. (2015) quantitatively estimated that this radius increases with distance from the Sahara: from ∼13.7 microns (at longitude ∼27oW) to ∼17.2 microns (at longitude ∼48oW). This can be explained by the decrease in CCN numbers associated with the decreasing numbers of Saharan dust particles with distance from the Sahara.

References Kishcha P., da Silva A., Starobinets B., Long C.N., Kalashnikova O., Alpert P. (2014). Meridional distribution of aerosol optical thickness over the tropical Atlantic Ocean. Atmospheric Chemistry and Physics Discussion 14, 23309-23339, doi:10.5194/acpd-14-23309-2014.

Kishcha P., da Silva A., Starobinets B., Long C.N., Kalashnikova O., Alpert P. (2015). Saharan dust as a causal factor of hemispheric asymmetry in aerosols and cloud cover over the tropical Atlantic Ocean. International Journal of Remote Sensing 36, 3423-3445, doi: 10.1080/01431161.2015.1060646.

---

## Short Comment (SC2) · 16 Nov 2016

Large uncertainty of the estimated dust radiative effect in winter and the contribution of non-dusty aerosols.

Based on satellite-retrieved parameters of cloud properties, the authors (Amiri-Farahani et al., 2016) concluded that, in the winter season, the dust – cloud radiative effect is "weakly positive $0.92 \pm 2.86$ W/m2". However, in fact, their estimate indicates that, in winter, the dust – cloud radiative effect could be either positive or negative, because of the large uncertainty of their estimate. The presence of non-dusty aerosols could also be a causal factor for the above-mentioned large uncertainty. In our papers (Kishcha et al., 2014, 2015), using NASA MERRAero reanalysis, we showed that, in

winter, Saharan dust is not the predominant aerosol species over the tropical North Atlantic, including the area 45W – 15E; 0N – 35N under the study by Amiri-Farahani et al. (2016). Apart from dust, non-dusty aerosols, such as carbonates (organic and black carbon), sea salt and sulfates also significantly contribute to the total AOD over the tropical North Atlantic. As shown in Fig. 1 (below), in contrast to July, in January dust dominates other aerosol species only near the African coast. The non-dusty aerosol species could affect cloudiness in a different manner. In the winter season, absorbing aerosols, such as organic and black carbon, produce mainly a positive semi-direct radiative effect, similar to the dust effect. Sulfates and sea salt, non-absorbing aerosols, produce a negative indirect radiative effect, acting as effective CCN. Thus, non-dusty aerosols, producing either positive or negative radiative effects, significantly contribute to the large uncertainty of the aerosol-cloud radiative effect in the winter season.

References: Kishcha P., da Silva A., Starobinets B., Long C.N., Kalashnikova O., Alpert P. (2014). Meridional distribution of aerosol optical thickness over the tropical Atlantic Ocean. Atmospheric Chemistry and Physics Discussion 14, 23309-23339, doi:10.5194/acpd-14-23309-2014.

Kishcha P., da Silva A., Starobinets B., Long C.N., Kalashnikova O., Alpert P. (2015). Saharan dust as a causal factor of hemispheric asymmetry in aerosols and cloud cover over the tropical Atlantic Ocean. International Journal of Remote Sensing 36, 3423-3445, doi: 10.1080/01431161.2015.1060646.

———————————————————

[Figure]

**January**

40N

EQ

40S

**July**

40N

EQ

40S

**Ratio**

0.1  0.2  0.5  1  1.5  2

**Fig. 1.** The ratio of dust aerosol optical depth (AOD) to AOD of all other aerosol species, based on the 10-year MERRAero data (July 2002 - June 2012).

---

## Referee Comment (RC1) · W. Wang (Referee) · 25 Nov 2016

Page 2 line 24, Sc have a small effect on outgoing longwave radiation. Give some references.

Page 7, line 20 to 23. You estimate dust-cloud radiative effect by using the data where both dust and MSc exist. However, since dust and cloud possibly distribute at different height, dust may have little or ignore effect on clouds (such as your results in Fig. 10 ). Wang et al. (2010) define dusty clouds (the height difference between dust and cloud less than 50 m) to study dust effects on clouds. The height differences between dust and MSc should also be given here.

[Figure]

The dust-cloud radiative effect could be either positive or negative by method 1 during Winter, Spring and Fall, but the RF is negative during Summer from the results in Table 2. Since the sign of RF is affected by the height of dust (Huang et al., 2014), the vertical profile of dust in spring and autumn should also be given and discussed.

Fig. 10. The author conclude SDE is the dominant effect. However, the results in Fig.10 which also represents the semi-direct effect of dust is not really obvious. The authors should give the significance degree for Fig 10.

References: (1) Huang, J., T. Wang, W. Wang, Z. Li, and H. Yan. Climate effects of dust aerosols over East Asian arid and semiarid regions, Journal of Geophysical Research: Atmospheres, 119, 11398–11416, doi:10.1002/2014JD021796, 2014. (2) Wang, W., J. Huang, P. Minnis, Y. Hu, J. Li, Z. Huang, J. Ayers, and T. Wang, Dusty cloud properties and radiative forcing over dust source and downwind regions derived from A-Train data during the Pacific Dust Experiment, Journal of Geophysical Research, 115 (2010), D00H35, doi:10.1029/2010JD014109.

---

## Referee Comment (RC2) · Anonymous Referee #2 · 7 Mar 2017

The radiative impact of Saharan dust on North Atlantic stratocumulus clouds (SCs) was estimated in this study by integrating A-train satellite constellation observations, MACC reanalysis and ERA-interim data together. The first indirect effect and combination of the second indirect effect and semi-direct effect were also quantified following the methods of Quass et al. (2008) and Chen et al. (2014). It is interesting to study dust aerosol effect on cloud according to observation and theoretical assumption. However, there are still some specific questions and methods should be addressed.

1. P1, line 7, the authors conclude that two methods yield similar results for the annual mean aerosol-cloud radiative effect. Actually, there is a big difference in standard deviation except for similar mean value.

[Figure]

2. P1, Line 17, "Semi-direct effect can result in negative (absorbing aerosol lies above low clouds) and positive (absorbing aerosol lies within low clouds)". It is confused that how to tell from the aerosol layer position above or within low clouds. As shown the author's statistic results, the 50-90

3. P4, Line 16, the same question as above, how did the authors quantify how much dust is within or even below clouds using CALIPSO? According to my understanding, most aerosols within and below clouds cannot be detected by CALIOP. If the clouds with cloud optical thickness less than 4 are excluded as described by authors, the aerosols within and below clouds will never be detected.

4. Suggest to give reader much more specific explanation about how to select SCs in the study area. More reader as me will confuse the connection between marine SCs regime and CERES cloud properties. Because authors told us that the SCs regime are defined only according to vertical velocity and LTS from ERA-Interim data. The definition exactly increase the convenient of selecting SCs, but we cannot understand how to obtain SCs cloud properties from CERES data and how to screen the effect of ice cloud at multilayer cloud system in this study. Because the calculation of planetary albedo according to function (2) ignores the contribution of ice clouds. Despite SCs are warm clouds, the ice cloud above SCs should be also screened according to such as cloud top pressure or cloud top temperature and so on.

5. P2, Line 5, please refer the following paper about dust semi-direct effect. Huang, J., P. Minnis, B. Lin, T. Wang, Y. Yi, Y. Hu, S. Sun-Mack, and K. Ayers, Possible influences of Asian dust aerosols on cloud properties and radiative forcing observed from MODIS and CERES, Geophysical Research Letters, 33 (6) (2006), L06824, doi:10.1029/2005GL024724.

---

## Author Comment (AC1) · 15 Apr 2017

Response: Thanks for this short comment.

This part is added to the introduction part: Kishcha et al., (2015) focussed on the tropical Atlantic Ocean (30° N–30° S). They find that during a 10-year study period (July 2002–June 2012), in July, dust intrusions from the Sahara into the tropical Atlantic contribute to significant cloud cover up to 0.8 – 0.9 along Saharan Air Layer (SAL). They suggest that the increase in cloud cover could be explained by the formation of shallow stratocumulus clouds below the temperature inversion with the assistance of settling Saharan dust particles.

---

## Author Comment (AC2) · 15 Apr 2017

Thanks for this useful comment.

This part is added to the result section: The aerosol-cloud radiative effect is weakly positive during boreal winter. The presence of non-dusty aerosols could also be a reason of the large uncertainty. Kishcha et al., (2015) show that, in winter, Saharan dust is not the predominant aerosol species over our study area. In winter non-dusty aerosols, such as carbonates (organic and black carbon), sea salt and sulfates also significantly contribute to the total AOD over the tropical North Atlantic. Absorbing aerosols, such as organic and black carbon, produce mainly a positive semi-direct radiative effect, similar to the dust effect. Sulfates and sea salt, non-absorbing aerosols,

produce a negative indirect radiative effect, acting as effective CCN. Thus, non-dusty aerosols, producing either positive or negative radiative effects, significantly contribute to the large uncertainty of the aerosol-cloud radiative effect in winter.

---

## Author Comment (AC3) · 15 Apr 2017

We would like to thank the reviewer for his/her constructive comments. The responses to the referee is formatted as follows: The original comments are given in black The author's response is given in red The changes in manuscript are given in green

Review by Referee #1 (W. Wang):

1st comment: Page 2 line 24, Sc have a small effect on outgoing longwave radiation. Give some references.

**Response:**

Two references are added to the manuscript.

**The manuscript has been changed as follows:**

Stratocumuli strongly reflect incoming solar radiation (Chen et al. 2000) and exert only a small effect on the outgoing longwave radiation. Overall, they exert a strong negative net radiative effect that markedly affects Earth's radiative balance (e.g., Stephens and Greenwald 1991; Hartmann et al. 1992).

Hartmann, D. L., M. E. Ockert-Bell, and M. L. Michelsen, 1992: The effect of cloud type on earth's energy balance—Global analysis. J. Climate, 5, 1281–1304.

Stephens, G. L., and T. J. Greenwald, 1991: Observations of the Earth's radiation budget in relation to atmospheric hydrology. Part II: Cloud effects and cloud feedback. J. Geophys. Res., 96, 15 325–15 340.

**2nd comment:**

Page 7, line 20 to 23. You estimate dust-cloud radiative effect by using the data where both dust and MSc exist. However, since dust and cloud possibly distribute at different height, dust may have little or ignore effect on clouds (such as your results in Fig. 10). Wang et al. (2010) define dusty clouds (the height difference between dust and cloud less than 50 m) to study dust effects on clouds. The height differences between dust and MSc should also be given here.

**Response:**

We agree that it would be interesting to separate the analysis to whether the dust is below, at the same height or above the clouds but dust aerosol optical depth (DAOD) from CALIPSO is available only two or three days a month and the DAOD from MACC is not vertically resolved. Dust must not necessarily be at the same height as the clouds to have an effect on clouds. Several studies (e.g. Koch and Del Genio, 2010; Wilcox, 2010; Constantino and Bréon, 2013) have shown that absorbing aerosol, which is above the clouds, can have an influence on the clouds below e.g. Wilcox (2010) estimates SDE where layers identified as cloud features occur predominantly below 1.5 km and features identified as layers of aerosol occur predominantly between 2 km and 4 km. The statistical method used in our study allows assessing the effect of a vertically integrated variable like DAOD on stratocumulus clouds, for example Chen et al. (2014) estimate global aerosol-cloud radiative forcing for marine warm clouds without any assumption for height difference between clouds and aerosols. Most of the dust in summer is indeed above the clouds in the studied region (Fig. 8c and 10c), but for other seasons the opposite was found. Summer is the season when the total radiative effect of dust is largest (Table 2) therefore dust seems to have an influence on the stratocumulus clouds even when it is above them.

The usage of a  $N_d$  mediated cloud fraction sensitivity to AI not only suppresses the impact of meteorological covariations on this sensitivity but also to the effect of absorbing aerosol on cloud fraction. We added this to the manuscript:

Figure 10 shows that the sensitivity of cloud fraction to AI is relatively weak. It also shows that this sensitivity is positive (negative) during summer (winter) for most of the study area, which shows that cloud fraction increases (decreases) when AI increases. Using equation (8) suppresses next to meteorological covariations also part of the effect of absorbing aerosol on cloud fraction i.e. the sensitivity in Figure 10 is a conservative estimate.

Wang et al. (2010) is a relevant paper to our work, thus this part is added to the introduction part:

Wang et al. (2010) compare dusty and pure cloud properties and radiative forcing over northwestern China (source region) and over the northwestern Pacific (downwind region). Dusty clouds are defined as clouds that extend into a dust plume environment (i.e., dust aerosols observed within 50 m of the cloud), while pure clouds are clouds having no dust aerosols within 500 m around them. They show that dust aerosols change the microphysical characteristics of clouds, reducing the cloud effective particle size and, possibly, cloud optical depth, LWP, and ice water path (IWP). They show that dust aerosols cause an instantaneous net cooling effect in the source and downwind regions respectively.

**3rd Comment:**

The dust-cloud radiative effect could be either positive or negative by method 1 during Winter, Spring and Fall, but the RF is negative during Summer from the results in Table 2. Since the sign of RF is affected by the height of dust (Huang et al., 2014), the vertical profile of dust in spring and autumn should also be given and discussed.

The manuscript has been modified (both in the text and abstract) as follows: Since the sign of the dust-cloud radiative effect is affected by the height of dust (Huang et al., 2014), to investigate the role of the SDE over the region, we look at the vertical profile of Saharan dust from CALIPSO. Figure 8 shows that during winter, most of the dust burden resides between 0-1 km. In contrast, during spring there are two peaks in Saharan dust: one peak is within the marine boundary layer (between 0-1km), and the rest resides above the boundary layer., with the peak above boundary layer being smaller than that within the boundary layer. During summer, similar to spring, there are two peaks, but most of dust resides above the boundary layer. During fall the amount of dust is less than in other seasons and most of dust burden resides between 0-1 km, with some dust between 1-4 km. The horizontal solid and dashed red lines in Figure 8 are average CERES MSc cloud top heights  $\pm$  one-sigma respectively for each season. The average cloud top heights in summer and spring are lowest with 1.9 $\pm$ 0.4 km respectively 2.0 $\pm$ 0.4 km, and highest in winter and fall with 2.2 $\pm$ 0.3 km as shown in Figure 9. CALIPSO shows that 88.3%  $\pm$ 8.5% of dust resides below 1.5 km in winter. During summer, however, there are two peaks, with 35.6%  $\pm$ 13% below 1.5 km and 44.4%  $\pm$ 9.2% between 2 and 4 km.

Figure 8. Vertical profile of dust from CALIPSO in (a) winter, (b) spring, (c) summer, (d) fall. Solid and dashed red lines show CERES MSc cloud top height ± one-sigma respectively for each season.

---

## Author Comment (AC4) · 15 Apr 2017

We would like to thank the reviewer for his/her constructive comments. The responses to the referee are formatted as follows:

The original comments are given in black

The author's response is given in red

The changes in manuscript are given in green

Review by Anonymous Referee #2

1st comment:
P1, line 7, the authors conclude that two methods yield similar results for the annual mean aerosol-cloud radiative effect. Actually, there is a big difference in standard deviation except for similar mean value.

Response:
Thanks for this comment; we changed our method to calculate the annual radiative effect. Now, the seasonally weighted mean radiative effect is calculated and uncertainties are calculated based on the propagation of uncertainty. In this new method, the annual radiative effect for Method 1 (Method 2) is $-1.5\pm1.4$ ($-1.5\pm1.6$) W/m$^2$. Originally we calculated the annual radiative effect by putting all data together and calculated the linear regressions of each partial derivative. Because of missing values, there was a mistake in the calculation of the uncertainties in Method 2, therefore the uncertainties have been changed as follows:
Winter: $-0.6\pm1$ W/m$^2$, Spring: $-1.3\pm4$ W/m$^2$, Summer: $-4.3\pm4.1$ W/m$^2$, Fall: $-1\pm2.5$ W/m$^2$

2nd comment:
P1, Line 17, "Semi-direct effect can result in negative (absorbing aerosol lies above low clouds) and positive (absorbing aerosol lies within low clouds)". It is confused that how to tell from the aerosol layer position above or within low clouds. As shown the author's statistic results, the 50-90

Response:
The statistical method used in our study allows assessing the effect of a vertically integrated variable like DAOD on stratocumulus clouds so for the calculation of the aerosol-cloud radiative effect it is not necessary to know the position of the aerosol layer relative to the low clouds (see also the response to the 2nd comment in the review by W. Wang).

The manuscript has been modified as follows (both in abstract and text parts):
The persistent MSc are low and confined within the boundary layer. CALIPSO shows that 61.8% $\pm12.6$ of Saharan dust resides above North Atlantic MSc during summer for our study area. This is consistent with a relatively weak first aerosol indirect effect, and also suggests the second aerosol indirect effect plus semi-direct effect (the second term in Method 1) is dominated by the semi-direct effect. In contrast, the percentage of Saharan dust above North Atlantic MSc in winter is 11.9% $\pm10.9$ which is much lower than in summer. CALIPSO also shows that 78% $\pm12.4$ of the dust resides below 1.5 km altitude

in winter. During summer, however, there are two peaks, with 31.1% ±12.9 below 1.5 km and 44.4% ±9.2 between 2 and 4 km

Kok et al. (2017) show that the dust found in the atmosphere is substantially coarser than represented in current global climate models. As coarse dust warms the climate, the temperature inversion is stronger and yields thickening of the underlying clouds.

3$^{rd}$ comment:
P4, Line 16, the same question as above, how did the authors quantify how much dust is within or even below clouds using CALIPSO? According to my understanding, most aerosols within and below clouds cannot be detected by CALIOP. If the clouds with cloud optical thickness less than 4 are excluded as described by authors, the aerosols within and below clouds will never be detected.

Response:
We got the data from Amiridis et al. (2013) and the product utilizes cloud-free CALIPSO profiles. In any case they could never retrieve the amount of dust below or within a cloud. Clouds could have been close to these profiles (horizontally), but not in the same columns that are used to provide dust properties. We use CERES cloud top heights and dust extinction coefficients from CALIPSO in 1°x1° grid boxes to calculate the percent of dust above clouds.

This part is added to the manuscript:
The extinction coefficient of dust for each level is obtained from CALIPSO and vertically integrated to calculate DAOD for each grid box, and then extinction coefficients above the CERES cloud top heights are vertically integrated and divided by DAOD to give the percent of dust above the clouds. The computation is done on a 1°x1° grid.

4$^{th}$ comment:
Suggest to give reader much more specific explanation about how to select SCs in the study area. More reader as me will confuse the connection between marine SCs regime and CERES cloud properties. Because authors told us that the SCs regime are defined only according to vertical velocity and LTS from ERA-Interim data. The definition exactly increase the convenient of selecting SCs, but we cannot understand how to obtain SCs cloud properties from CERES data and how to screen the effect of ice cloud at multilayer cloud system in this study. Because the calculation of planetary albedo according to function (2) ignores the contribution of ice clouds. Despite SCs are warm clouds, the ice cloud above SCs should be also screened according to such as cloud top pressure or cloud top temperature and so on.

Response:
Only those grid points and days are selected in our analysis where 500 hPa vertical

velocity > 10 hPa day−1 and LTS > 18.55 K (see Figure 2). The contribution of ice clouds in planetary albedo in our study area is small and negligible, but we also remove them. Clouds with cloud top pressure less than 500 hPa and cloud top temperature below 270 K are screened out (This part will be added to the manuscript). On page 7, line 4 we mention: where f is the marine stratocumulus cloud coverage and clouds are not obscured by overlying ice clouds.

This part is added to the manuscript:
… a LTS criterion is used, defined as LTS = $\theta_{700hPa} - \theta_{1000hPa}$ > 18.55K (where $\theta$ is the potential temperature). Only grid points and days within the MSc regime are used in the analysis.

and:
…where f is the marine stratocumulus cloud coverage and clouds are not obscured by overlying ice clouds (i.e. the small number of scenes with ice clouds in our study area are removed from the analysis).

5[th] comments:
P2, Line 5, please refer the following paper about dust semi-direct effect. Huang, J., P. Minnis, B. Lin, T. Wang, Y. Yi, Y. Hu, S. Sun-Mack, and K. Ayers, Possible influences of Asian dust aerosols on cloud properties and radiative forcing observed from MODIS and CERES, Geophysical Research Letters, 33 (6) (2006), L06824, doi:10.1029/2005GL024724.

Response:
The reference is added to the manuscript.

The paper is added to introduction part:
Huang et al. (2006) analyzed the effect of dust storms on cloud properties and radiative forcing over Northwestern China from April 2001 to June 2004. Due to changes in cloud microphysics, the instantaneous net radiative forcing is increased from -161.6 W/m$^2$ for dust-free clouds to -118.6 W/m$^2$ for dust contaminated clouds.